# Plant Growth and Yield Response to Salinity Stress of Rice Grown under the Application of Different Nitrogen Levels and *Bacillus pumilus* Strain TUAT-1

**Khin Thuzar Win** [1,*,†] , **Aung Zaw Oo** [1,‡] **and Tadashi Yokoyama** [1,2]

[1] The Faculty of Agriculture, Tokyo University of Agriculture and Technology, Saiwaicho 3-5-8, Fuchu, Tokyo 183-8509, Japan

[2] The Faculty of Food and Agricultural Science, Fukushima University, Kanayagawa, Fukushima City 960-1296, Japan

\* Correspondence: kint688@affrc.go.jp; Tel.: +81-298388377

† Current address: Institute of Agrobiological Sciences, National Agriculture and Food Research Organization, 3-1-3 Kanondai, Tsukuba 305-8604, Japan.

‡ Current address: Japan International Research Center for Agricultural Sciences, 1-1 Ohwashi, Tsukuba 305-8686, Japan.

**Abstract:** Rice is an important food crop, and its production is significantly affected by salt stress under the changing climate. Soil-inhabiting microbial inoculants as well as efficient nitrogen (N) nutrition may have ameliorative effects on rice growth and yield under salt stress. However, the effects of the interaction between N application and microbial inoculants on the growth and grain yield of rice under salt stress is not yet fully understood. This study aimed to clarify whether the use of the *Bacillus pumilus* strain TUAT-1 biofertilizer, along with the right amount of N fertilizer, would alleviate salt stress in lowland rice production. We applied the *Bacillus pumilus* strain TUAT-1 as a biofertilizer in combination with different rates of N fertilizer: control (0% N), 2.64 g $(NH_4)_2SO_4$ per nursery tray (100% N: the farmer-recommended amount), and 3.96 g $(NH_4)_2SO_4$ per nursery tray (150% N). Salinity (100 mM of NaCl) was applied at the heading stage of rice plants in pot culture in the greenhouse, and the growth and yield components were accessed at harvest. In the nursery phase, the application of the biofertilizer TUAT-1 significantly increased seedling vigor and the root development of 21-day-old seedlings. Salinity stress at the heading stage significantly reduced chlorophyll content, panicle number, straw biomass, and grain yield; however, either the application of N alone or in combination with TUAT-1 ameliorated the salinity-related reduction in grain yield and yield component parameters. Plants receiving a high amount of N fertilizer (150% N) showed similar straw biomass and grain yield with or without TUAT-1 inoculation, regardless of saline or non-saline conditions. In both saline and control conditions, straw biomass and grain yield were higher in the plants inoculated with TUAT-1 than in those without TUAT-1. Specifically, the combined application of TUAT-1 and the farmer-recommended N level (100% N) led to an increase of 8% in straw biomass and 15% in grain yield under saline stress when compared with their respective plants without TUAT-1. Straw biomass and grain yield were similar in the (un-inoculated) plants grown under normal conditions and TUAT-1 + 100% N under salinity treatments, because TUAT-1 enhanced root development, which may promote soil nutrient uptake. Our results indicated that combined nursery application of TUAT-1 biofertilizer and 100% N fertilizer rate has the potential to boost the capacity of this bacteria to increase seedling vigor, which subsequently ameliorated the salt-induced reduction in the grain and straw yield.

**Keywords:** grain yield; nitrogen fertilizer; plant growth-promoting bacteria; rice seedling vigor; salt stress

## 1. Introduction

Rice (*Oryza sativa* L.) is an important cereal crop, and more than half of the world's population depends on it [1]. However, rice is highly salt sensitive relative to other major cereals, and salinity problems are the major limiting factor in rice production. Soil salinity is becoming a global problem in the face of the changing climate. The land area affected by salinity is constantly increasing. More than 800 million hectares of land worldwide are currently affected by salt concentrations that could significantly impact crop yields [2]. Therefore, several problems caused by salt stress have prompted researchers to investigate the improvement of salt tolerance in agricultural crops with the aim of promoting crop productivity in an environmentally friendly manner [3].

In general, growth reduction due to salinity is attributed to an impact on osmosis due to lower water availability in short-term stress, and long-term salinity impacts ion toxicity due to imbalanced plant nutrient uptake and mineral nutrient status in plant tissues. Previous studies showed that excess salt in the soil or the nutrient solution led to a decrease in the N uptake of plants such as beans [4], barley [5], and rice [6].

Nitrogen fertilization in crops growing in arid climates can increase the salt tolerance of plants [7]. Efficient N nutrition can help crops cope with salt stress by minimizing the toxic effects, and thus, efficient nutrient management would lead to resilience to salt stress [8]. In addition, an apparent increase in salinity tolerance has been noted when N fertilizer is applied beyond the optimal rate for normal field conditions [9], indicating that increased fertilization, particularly N, can mitigate the deleterious effects of salinity [10]. Mohanty et al. [11] reported that increased N fertilization could alleviate the inhibiting effects of salinity on rice plants to some extent. However, excessive N fertilization will lead to inefficient use of N and significant N losses to the environment, negative impacts on air and water quality, biodiversity, ecological impacts, and human health [12].

As an alternative, using plant growth-promoting bacteria (PGPB) is one of the environmentally friendly approaches to improve the salt tolerance of crops [13]. Numerous studies reported that several genera of bacteria like *Pseudomonas, Bacillus, Pantoea*, and *Burkholderia* were beneficial for providing resistance in peas, maize, wheat, grapevine, and common bean against various abiotic stresses, including drought and salt stress [14–17].

Recent findings suggest that PGPB is a promising alternative to alleviate salt stress [18–20], and the use of microbes is developing an important role in the management of biotic and abiotic stresses. The beneficial microbes inhabit the plant rhizosphere and stimulate plant growth via several indirect and direct mechanisms [21], such as reducing stress ethylene production [22], lowering the uptake of sodium [23] and production of IAA and siderophore [24], and increasing K+ content and proline level [25]. Sirajuddin et al. [26] reported that inoculation with *Bacillus pumilus* conferred salt stress resistance to the rice plants.

Previous studies indicated that the *Bacillus pumilus* strain TUAT-1 increases the growth of forage rice (Leaf Star cultivar) [27], regular rice (Koshihikari cultivar) [28], and vegetables [29], mainly due to enhanced root development associated with increased nutrient uptake by plants. However, although a large body of data indicates the growth enhancement of TUAT1, the bacteria–plant interactions under saline stress combined with N fertilization in rice remain concealed.

In the young seedling and reproductive stages, rice plants are sensitive to salt stress [30], but they are relatively tolerant at seed germination [31]. Although salinity can affect all plant growth stages, the effect was more pronounced at the reproductive stage due to reduced grain yield; therefore, the resultant salt tolerance in the reproductive stages was considered more useful [32]. Understanding the interactions between N fertilization and TUAT-1 is of great importance for improving rice growth and grain yield under saline stress. Our hypothesis was to address the additive effects of N fertilization and TUAT-1 on the enhancement of the growth and grain yield of rice under saline stress at the reproductive stage. The main objective was to study the combined effects of the *Bacillus pumilus* strain TUAT-1 biofertilizer and different N application rates on rice growth and grain yield in

the presence and absence of salt stress. We hypothesized that using the *Bacillus pumilus* strain TUAT-1 along with the right amount of N fertilizer would alleviate salt stress in lowland rice production, and that it could be used as a potential biocontrol for areas affected by salinity.

## 2. Materials and Methods

### 2.1. Nursery Preparation

Koshihikari, a dominant rice (*Oryza sativa* L.) variety in Japan, was used in this study. In order to eliminate possible contamination, rice seeds were surface sterilized with 70% alcohol for 30 s, followed by immersion in 2% sodium hypochlorite for 2 min. Next, the seeds were washed five times with sterile distilled water and pre-germinated in sterilized distilled water at 27 °C for two days. Before sowing pre-germinated seeds, the nursery soils were mixed with biofertilizer (5% zeolite and silica gel-coated granular bio-inoculant with or without the *Bacillus pumilus* strain TUAT-1 strain) at a rate of 5 g biofertilizer per 100 g soil (160 g in 3.2 kg soil) according to the protocol of Win et al. [33]. The TUAT-1 *Bacillus* cell density in the biofertilizer was about $1.2 \times 10^7$ colony-forming units (CFU) $g^{-1}$.

Each nursery tray (30 cm × 60 cm) was filled with 3.2 kg of commercially available nursery soil (Shinano Soil, Shinano Baiyoudo Co., Ltd., Nagano, Japan). Then, pregerminated seeds were evenly sown in the nursery tray. In the nursery experiment, TUAT-1 biofertilizer was factorially combined with different rates of N fertilization (1) 0% N, (2) TUAT-1 + 0% N, (3) 100% N, (4) TUAT-1 + 100% N, (5) 150% N, and (6) TUAT-1 + 150% N. The 100% N rate consisted of 2.64 g $(NH_4)_2SO_4$ (ammonium sulfate) per nursery tray (recommended practice for rice farmers in Japan), whereas no nitrogen fertilizer was applied for 0% N treatment, and 3.96 g $(NH_4)_2SO_4$ was used for 150% N treatment.

The TUAT-1 isolate was grown as a liquid culture in Trypticase Soy Broth (TSB) (Becton Dickinson, Sparks, MD, USA). The overnight cultures were centrifuged at 10,000 rpm for 10 min at 6 °C. They were then washed twice in sterile milli-Q water and diluted to an optical density of 0.4 at 600 nm, corresponding to approximately $10^7$ cells $mL^{-1}$. Next, 500 mL was applied to each nursery tray one week after seed sowing to ensure root colonization. Rice seedlings were grown under greenhouse conditions for 21 days.

### 2.2. Nursery Seedling Vigor Measurement

The 21-day-old seedlings were randomly selected from four locations in each nursery tray. The following data were recorded from 30 seedlings: shoot length, shoot and root fresh and dry weight, and root surface area. For root analysis, roots were gently washed with deionized water, and the root surface area and total root length were measured with an image analyzer (Win-Rhizo REG V 2004 b; Regent Inc., Quebec, QC, Canada).

### 2.3. Pot Experiments and Plant Growth Conditions

The pot experiments were carried out inside a greenhouse located at the Tokyo University of Agriculture and Technology. Greenhouse settings were: day/night mean temperature of 30/25 °C, relative humidity of 63–85%, an average of a 12 h photoperiod, and an average maximum photosynthetically active radiation (PAR) of 500 μmol photons $m^{-2}s^{-1}$ (190SA quantum sensor, LI-COR). Commercial Shinano soils were heat sterilized at 121 °C for 1 h on three successive days. Twenty-one days after nursery raising, the above-treated seedlings were transplanted with three seedlings per pot (20 cm in diameter) filled with 3.5 kg of sterilized soil. The pots were fertilized with three different rates of N as in nursery seedling treatments: N: 0 g, 2.64 g, and 3.96 g of ammonium sulphate per pot, corresponding to 0%, 100%, and 150% N, respectively. All treatment combinations were factorially arranged in a randomized complete block design with four replications.

All pots were irrigated with tap water and maintained at a water level of 5 cm until salt treatment application. In order to study the effects of salt stress at the reproductive stage of the rice plants, salt (100 mM NaCl) was applied at the booting stage (about eight weeks after transplanting). At the beginning of the salinity treatment, the NaCl concentration

was gradually increased by 50 mM NaCl at 3-day intervals until the required salinity of 100 mM NaCl was reached. Next, each pot was irrigated once in 3 days with 100 mM NaCl or tap water (control), and a water level of 5 cm was maintained in the pot. The salt treatment continued until final drainage for harvest. Final drainage was carried out ten days before harvest, and the rice plants were harvested at the maturity stage (120 days after transplanting).

### 2.4. Growth and Yield Attributes

After 3 weeks of NaCl stress, the relative chlorophyll content of the flat leaf was measured using a SPAD (soil plant analysis development) analyzer (Minolta, by Hydro Agri, Dülmen, Germany), which measures the transmission of the wavelengths absorbed by chlorophyll in intact leaves (mid position). Each replicate was measured 30 times, and the mean was used for analysis. Plant height was measured by using a ruler from the ground level to the top of the longest leaf at the seed-filling stage. At harvest, the following data were recorded: number of panicles, straw yield, and grain yield. Grain yield was adjusted to 14% moisture content, while straw yield was recorded on an oven-dry (80 °C) basis.

### 2.5. Statistical Analysis

The data were analyzed by three-way analysis of variance (ANOVA) using STAR—Statistical Tool for Agricultural Research—version 7.0 software (International Rice Research Institute, Los Baños, Philippines). The treatment means were compared at a 5% level of probability using Turkey's HSD test.

## 3. Results and Discussions

### 3.1. Effect of N Levels and TUAT-1 on Seedling Vigor

The application of TUAT-1 bio-inoculant and N fertilizer to the nursery bed significantly increased the shoot length, shoot and root dry weight, and root surface area of rice seedlings (Figure 1 and Table 1). A significant interaction effect was observed from N levels and TUAT-1 bio-inoculant on the root parameters of rice seedlings, except for root dry weight (Table 1).

**Table 1.** Effects of *Bacillus pumilus* TUAT-1 inoculation and different rates of N on root morphology in nursery seedlings. Data were analyzed by two-way ANOVA with the factors of TUAT-1 (T) and nitrogen levels (N). Means followed by the same letter are not statistically significant according to Turkey's HSD test.

|  | N Levels (%) | Root Dry Weight (g plant$^{-1}$) | Root Surface Area (cm$^2$ plant$^{-1}$) | Total Root Length (cm plant$^{-1}$) | Root Number plant$^{-1}$ |
|---|---|---|---|---|---|
| Un-inoculated | N0 | 0.015 [bc] ± 0.001 | 12.67 [d] ± 2.7 | 240.28 [e] ± 32.4 | 9.66 [d] ± 0.9 |
|  | N100 | 0.021 [b] ± 0.001 | 20.74 [c] ± 3.5 | 340.86 [d] ± 48.0 | 13.32 [bc] ± 1.1 |
|  | N150 | 0.021 [b] ± 0.001 | 29.75 [b] ± 5.3 | 450.84 [c] ± 66.0 | 14.63 [b] ± 1.4 |
| TUAT-1 | N0 | 0.028 [ab] ± 0.001 | 20.02 [c] ± 3.6 | 342.41 [d] ± 47.0 | 11.14 [c] ± 0.9 |
|  | N100 | 0.026 [ab] ± 0.001 | 35.89 [ab] ± 0.7 | 582.66 [b] ± 42.4 | 15.61 [ab] ± 1.2 |
|  | N150 | 0.031 [a] ± 0.003 | 42.61 [a] ± 0.2 | 753.73 [a] ± 54 | 16.19 [a] ± 1.1 |
| | | Analysis of variance (*p* value) | | | |
| N | | 0.000 | 0.000 | 0.015 | 0.010 |
| T | | 0.000 | 0.000 | 0.000 | 0.015 |
| N × T | | 0.651 | 0.028 | 0.015 | 0.015 |

Data are presented as mean ± standard deviation.

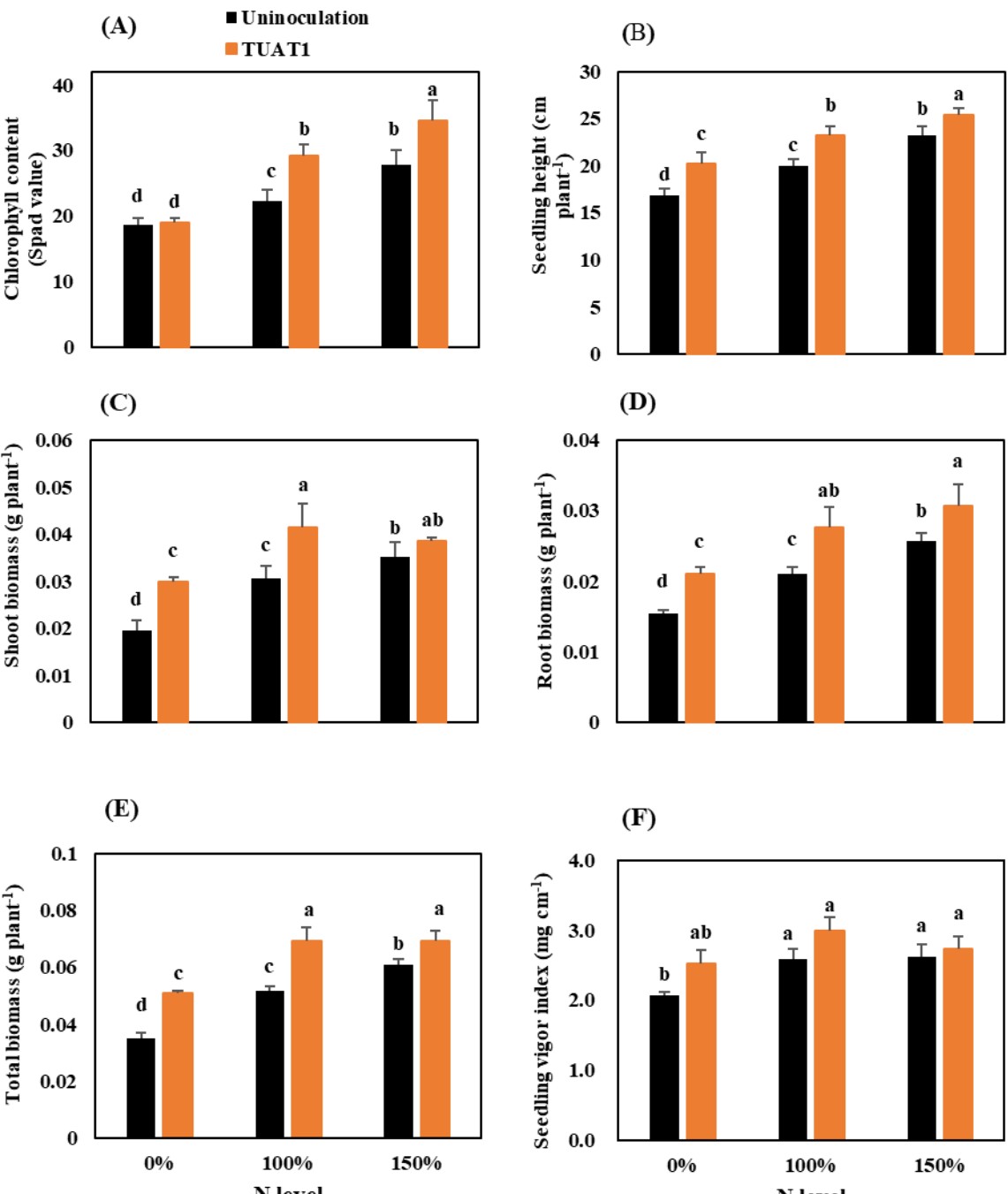

**Figure 1.** Combined effects of *Bacillus pumilus* TUAT-1 inoculation, and different rates of N on the (**A**) leaf chlorophyll contents (SPAD value), (**B**) seedling height, (**C**) shoot biomass, (**D**) root biomass, (**E**) total biomass, and (**F**) seedling vigor index of rice seedlings at 21 d after inoculation. Error bars represent standard error. The same letters indicate no significant differences among the treatments according to Turkey's HSD test.

Nursery application of TUAT-1 combined with different rates of N significantly increased the seedling height, shoot biomass, total biomass, and vigor index of seedling at 21 days after sowing. The growth promotion of TUAT-1 in total biomass was recorded as 46%, 34%, and 13% combined with N0, N100, and N150 treatments, respectively (Figure 1). The root parameters of rice seedlings changed with or without TUAT-1 at different rates of N application in the nursery. TUAT-1 significantly improved the root number and root length by increasing root surface area by 58%, 73%, and 43% with N0, N100, and N150

compared to their respective plants without TUAT-1. The increase in the root parameters plays a significant role in seedling vigor and plant growth promotion [34]. Seedling growth and seedling vigor are the most crucial growth phases of any crop, as they determine the amount of biomass produced. Particularly in rice, they are important for the development of tillers [35]. In addition, the root system plays a vital role in crop productivity, as the root absorbs essential nutrients from the soil and is a pathway to improved productivity and environmental outcomes [36].

Increased nutrient uptake by PGPB-inoculated plants was attributed to the production of plant growth regulators at the root interface, which stimulates root development and results in better uptake of water and nutrients from the soil [37]. The ability of TUAT-1 to improve plant growth by enhancing water and nutrient uptake has been widely reported [27–29]. In our previous investigation, the appropriate amount of N fertilization for the nursery bed with TUAT-1 biofertilizer resulted in significant growth effects on rice grain yields in the field study [28]. Our results indicated that the capacity of TUAT-1 presumably produces phytohormones as the main factor contributing to the ability of TUAT-1 to support root growth and improve nutrient and water uptake. These issues need to be further explored in the future.

### 3.2. Rice Growth and Yield under NaCl Stress

The effect of N levels and TUAT-1 on rice growth, grain yield, and yield components under salt stress are shown in Table 2. The ANOVA table showed that almost all of the growth and yield components significantly differed due to N levels. TUAT-1 treatment significantly enhanced the number of panicles, straw biomass, and grain yield. On the other hand, salinity significantly induced a reduction in growth and yield parameters. An interaction effect for grain yield was significant either between $N \times NaCl$ or $N \times NaCl \times TUAT-1$.

**Table 2.** Effects of *Bacillus pumilus* TUAT-1 inoculation and different rates of N on rice growth and grain yield. Data were analyzed by three-way ANOVA with the factors of TUAT-1 (T), N level (N), and salinity (S). N: Nitrogen level, T: TUAT-1, S: salinity.

| ANOVA (*p* Values) | | | | | |
|---|---|---|---|---|---|
| | SPAD Value | Plant Height (cm) | Number of Panicle | Straw Biomass (g plant$^{-1}$) | Grain Yield (g plant$^{-1}$) |
| N | 0.011 | 0.004 | 0.005 | 0.000 | 0.003 |
| T | 0.558 | 0.389 | 0.008 | 0.000 | 0.002 |
| N × T | 0.978 | 0.452 | 0.012 | 0.007 | 0.171 |
| S | 0.001 | 0.066 | 0.048 | 0.137 | 0.004 |
| N × S | 0.005 | 0.67 | 0.245 | 0.829 | 0.006 |
| T × S | 0.154 | 0.254 | 0.376 | 0.001 | 0.214 |
| N × S × T | 0.125 | 0.912 | 0.082 | 0.732 | 0.005 |

Growth is one of the most reliable indicators for assessing plant salt tolerance. Salinity treatments substantially reduced the chlorophyll content of flag leaf compared with their counterparts. However, its effect is smaller in the treatments of N150 with or without TUAT-1 (Figure 2B). Similarly, little or no difference in the yield and yield components was observed when plants were applied with N150 alone or in combination with TUAT-1 between salt and non-salt stress (Figure 2C,D and Figure 3). The reduction in grain yield was 20%, 10%, and 3% for N0, N100, and N150 treatments under saline stress compared to those without salt, respectively. An apparent decrease in the salt-induced reduction of grain yield was found when the N applied under saline conditions exceeded that of N0 treatment under non-saline conditions, implying that N fertilization may ameliorate the deleterious effects of salinity. In addition, the straw biomass and grain yield resulted in similar values in the (un-inoculated) plants grown under normal conditions and TUAT-1 + 100% N under saline conditions, as TUAT-1 enhanced root development, which may promote nutrient uptake from the soil. Likewise, Song et al. [38] proposed that N fertilization may be an

important agronomic approach to improve crop performance when exposed to saline conditions. This might be due to the accumulation of amino acids in the plant tissue [39], which is likely to balance the increased osmotic potential caused by NaCl stress, protecting the cell from damage and helping to improve physiological processes.

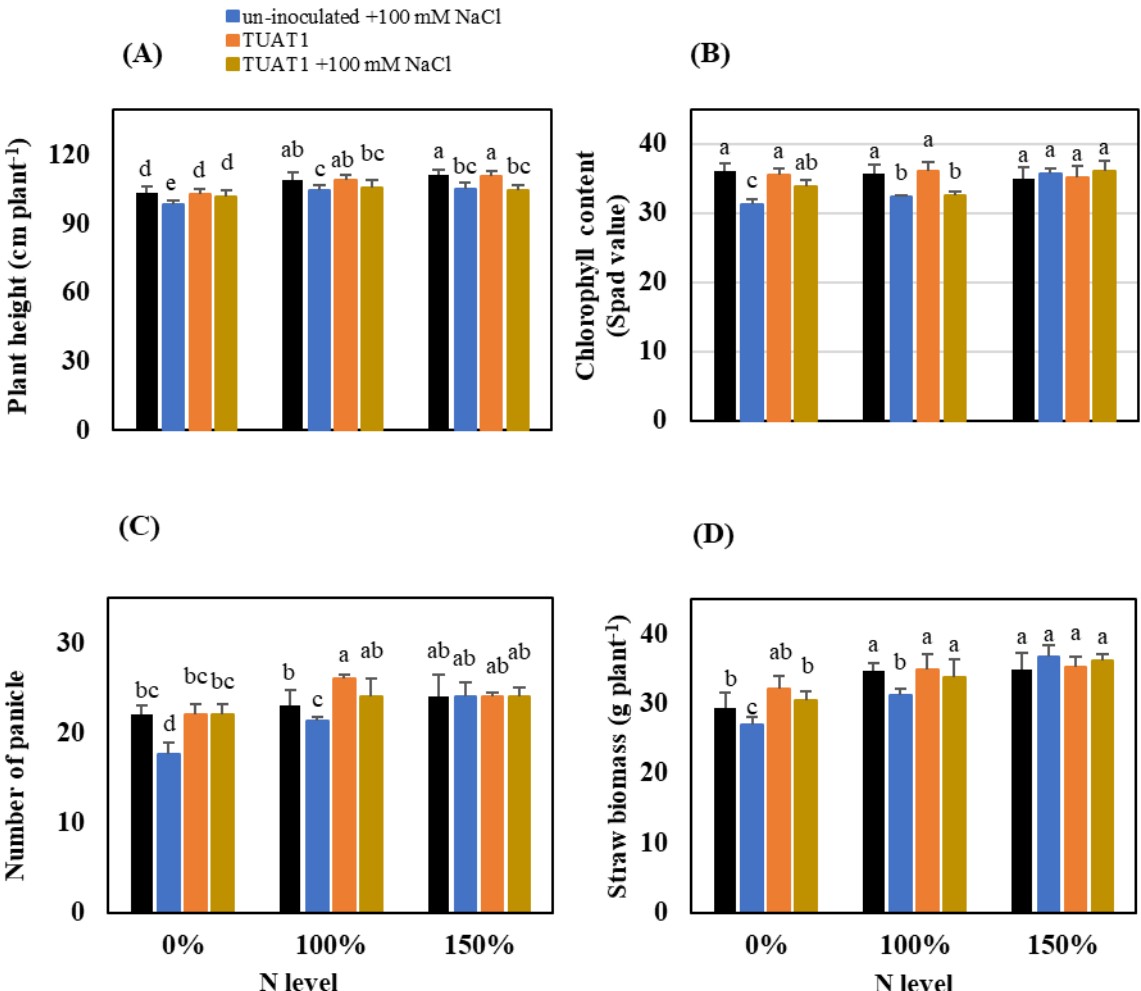

**Figure 2.** Combined effects of *Bacillus pumilus* TUAT-1 inoculation and different rates of N on the (**A**) plant height, (**B**) leaf chlorophyll contents (SPAD value), (**C**) number of panicles, and (**D**) straw biomass of rice plants. Error bars represent standard error. The same letters indicate no significant differences among the treatments according to Turkey's HSD test.

TUAT-1 inoculation with N0 treatment increased straw biomass by 9% and grain yield by 17% under the non-saline conditions compared to non-inoculation. Under saline stress, the grain yield of TUAT-1 + N0 was similar to that of N0 treatments without salinity. The application of TUAT-1 with N100 or N150 maintained its grain yield under the saline stress. A reduction in the grain yield and yield parameters of rice due to saline stress at the reproductive stage was reported by Razzaque et al. [40]. However, a higher rate of N fertilization under saline stress enhanced rice production [41]. Likewise, our results indicated that TUAT-1 inoculation with different N rates enhanced growth and grain yield under salt and non-salt conditions. Studies highlighted that stress tolerance is improved by PGPR through various mechanisms such as the production of gibberellins, indole acetic acid, cytokines, and some unidentified elements that result in increased root surface area, root length, root tips, and most importantly, increased nutrient content, thereby improving the health of the plant under stress [42,43].

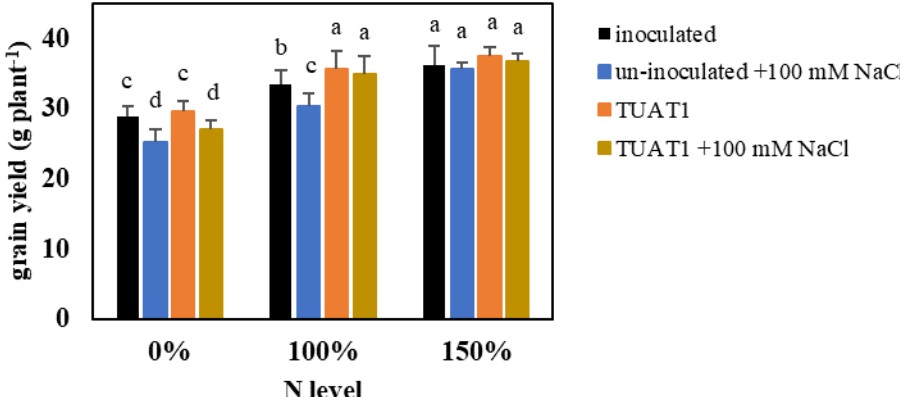

**Figure 3.** Combined effects of *Bacillus pumilus* TUAT-1 inoculation and different rates of N on the grain yield of rice. Error bars represent standard error. The same letters indicate no significant differences among the treatments according to Turkey's HSD test.

Okazaki and Yokoyama [44] reported that TUAT-1 encoded more than 25 sporulation-related genes, and that the genome also encodes genes involved in the synthesis of IAA and siderophores that promote plant growth. Furthermore, it is well known that cytokinin, as well as abscisic acid (ABA) accumulation and antioxidants produced by PGPR, can improve plant growth through ACC deaminase activity [45]. In the case of TUAT-1 bio-inoculant, however, such evidence is lacking. Therefore, based on our results, we proposed that the combined application of TUAT-1 and N fertilizer can be attributed to the increased root-to-shoot growth at the seedling stage (Table 1), which may affect the improvement of dry matter accumulation at the reproductive stage by supplying a sufficient amount of nutrients, water, and phytohormones to the shoots and subsequently ensuring an increase in rice productivity under salt stress conditions.

### 4. Conclusions

The results showed that the salt treatment imposed in the reproductive growth stage significantly reduced rice growth, biomass production, and grain yield. The application of *Bacillus pumilus* strain TUAT-1 to the nursery bed significantly improved seedling vigor and thus increased shoot biomass and grain yield under both saline and non-saline conditions. The application of higher rates of N also revealed the mitigation effect of salinity on plant growth and yield. Therefore, the *Bacillus pumilus* strain TUAT-1, along with the proper amount of N fertilizer application, would potentially alleviate saline stress in lowland rice production. For recommendation as a potential biofertilizer that enhances crop production and mitigates salt stress, further studies are warranted to elucidate the mechanism underlying the impact of N fertilization and TUAT-1 bio-inoculation on salinity tolerance in rice under real salt-affected field conditions.

**Author Contributions:** K.T.W.: experiment, sample analysis, data analysis, and writing manuscript. T.Y.: research idea, design, and data collection and analysis supervision. A.Z.O.: conceptualization, research idea, and data analysis. All authors proofread the final submission. All authors have read and agreed to the published version of the manuscript.

**Funding:** This research received no external funding.

**Institutional Review Board Statement:** Not applicable.

**Informed Consent Statement:** Not applicable.

**Data Availability Statement:** Not applicable.

**Acknowledgments:** This research was funded by MAFF Science and Technology Research Promotion Project for Agriculture, forestry and fisheries and food industries (26073C).

**Conflicts of Interest:** The authors declare no conflict of interest.

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
