# Peer review of "Plant Growth and Yield Response to Salinity Stress of Rice Grown under the Application of Different Nitrogen Levels and Bacillus pumilus Strain TUAT-1"

_2673-7655, doi:10.3390/crops2040031_

Round 1

Reviewer 1 Report (Previous Reviewer 1)

Study entitled “Plant growth and yield response to salinity stress of rice grown under the application of different N levels and Bacillus pumilus strain TUAT-1” contains novel information which can further strengthen the existing knowledge of the field. Scientists planned their study according to need of time. Moreover, the research area seems well motivated; data presented are sound; data analyses are technically correct and research findings support the claim and objectives properly made in the manuscript. Results are correctly presented and compared with existing knowledge of the field and I think results have some potential broader applicability. Please consider addressing following concerns and incorporate suggestions before any consideration to publish this work.

Minor Concerns

Line 27-28: 1. What do you mean, 0%, 100% and 150% N? It may be kg, make sure?

Line 41-43: Too lengthy sentence, rewrite please, how much increase or decrease, present in percentage.

Line 79: What is Tori and Win?

Line 141-147: Rewrite and add the instruments used for the measurement of various parameters.

Line 165: Micro nutrient status was not determined in the study.

Major Concerns

1.      Abstract lacks the importance of rice.

2.      Conclude all your results in a quantitative way in the abstract. Language improvement looks necessary as typo mistakes are observed while reviewing the draft.

3.      The basic effect of Na toxicity and nutrient deficiency was missed.

4.      Hypothesis and novelty of the study should be elaborated in more detail.

5.      Introduction It is directionless without synthesis of literature, hypothesis or study question.

6.      Directly state the results with significant findings. Make the results section concise and specific.

7.      Results It is poorly written. This needs to be re-written and get rectified from professional language editor and provide certificate from the company.

8.      Make the Discussion section separate.

9.      Try to discuss results with recent literature and provide reasoning of the responses recorded. The introduction and discussion section may be improved by citing the following recent findings;

 DOI: 10.35495/ajab.2020.05.273

DOI: 10.35495/ajab.2020.05.304

DOI: 10.35495/ajab.2020.05.295

DOI: 10.35495/ajab.2020.10.527

DOI: 10.35495/ajab.2019.06.273

DOI: 10.35495/ajab.2019.12.571

10.  Conclusion may be described with quantification and recommendation.

11. Use journal’s guidelines for the format of references within text and at the end.

Author Response

Reviewer1:

No.

Comments

Responses

1

Study entitled “Plant growth and yield response to salinity stress of rice grown under the application of different N levels and Bacillus pumilus strain TUAT-1” contains novel information which can further strengthen the existing knowledge of the field. Scientists planned their study according to need of time. Moreover, the research area seems well motivated; data presented are sound; data analyses are technically correct and research findings support the claim and objectives properly made in the manuscript. Results are correctly presented and compared with existing knowledge of the field and I think results have some potential broader applicability. Please consider addressing following concerns and incorporate suggestions before any consideration to publish this work.

Thank you very much for your constructive comments. We have followed all comments and have revised them accordingly.

2

Line 27-28: 1. What do you mean, 0%, 100% and 150% N? It may be kg, make sure?

We added up the amount of N fertilizer applied. See L28~32.

3

Line 41-43: Too lengthy sentence, rewrite please, how much increase or decrease, present in percentage.

We have shortened the sentenced accordingly and % increase is shown in L37. See L46~48.

4

Line 79: What is Tori and Win?

It was our mistake. We removed it. L91.

5

Line 141-147: Rewrite and add the instruments used for the measurement of various parameters.

We added the instrument when needed. L164.

6

Line 165: Micro nutrient status was not determined in the study.

Although we did not analyze micronutrient status in this study, we cited other related studies. To avoid confusion we removed it. L186-187.

7

1.      Abstract lacks the importance of rice.

We followed and revised. See L24-25.

8

2.      Conclude all your results in a quantitative way in the abstract. Language improvement looks necessary as typo mistakes are observed while reviewing the draft.

We have revised accordingly. See the abstract.

3.      The basic effect of Na toxicity and nutrient deficiency was missed.

In the abstract, we present their basic effect in L35~.

4.      Hypothesis and novelty of the study should be elaborated in more detail.

We added research aim and hypothesis. See L28~

9

5.      Introduction It is directionless without synthesis of literature, hypothesis or study question.

We have revised and included the hypothesis in the introduction. L67~72, L105~107

10

6.      Directly state the results with significant findings. Make the results section concise and specific.

We followed and revised accordingly. L175~

11

7.      Results It is poorly written. This needs to be re-written and get rectified from professional language editor and provide certificate from the company.

We followed and revised accordingly. We asked a native speaker to check the language and we could not make an effort with professional English editing services.

12

8.      Make the Discussion section separate.

The journal does not have strict formatting requirements; let us keep the results and discussion section parts together.

13

9.      Try to discuss results with recent literature and provide reasoning of the responses recorded. The introduction and discussion section may be improved by citing the following recent findings;

Thank you for providing the recent literature. We followed and revised accordingly. However, most of the literature provided is not directly related to the present study.

14

10.  Conclusion may be described with quantification and recommendation.

We highlighted to further confirm the results under real salt-affected field conditions for future application as biofertilizer. L281~282.

15

11. Use journal’s guidelines for the format of references within text and at the end.

We have formatted the references according to the guidelines.

Reviewer 2 Report (Previous Reviewer 2)

Review report on manuscript (crops-2003820) entitled: Plant growth and yield response to salinity stress of rice grown under the application of different N levels and Bacillus pumilus strain TUAT-1. Paper is well written, results are clearly presented and discussed. Following comments should be addressed before further process.   

 1)     The English language should be thoroughly checked. There are many wrongly written sentences and phrases. Below few corrections are mentioned:

a)      Line 49. space is required

b)      Line 79. The reference mention in two type, reference number and author name, author should use only reference number instead of author name in manuscript text.

c)      Line 99-102. Expression of sentence is not clear in material method section.

d)      Line 125.  Author should check unit of PAR of 500 mmol photons m-2s -1. It makes irrelevant data which is beyond the limit.

e)      Line 144. Each replicate was measured 30 times, it is an unrealistic approach.

f)       Line 219.  Figures, Tables and Schemes heading should be avoided while writing the manuscript. Author should delete it accordingly.

2)Author  used   abbrebiation N in title, but what it is denotining is not clear enough. Author should work on the  title such that it clearly mention the research work use of abrevation must be minimised  and should be explained properly in title.

3)Authors   should mention the final recommendation of different N levels and  Bacillus pumilus strain TUAT-1 on crop yield in abstract section of manuscript so that people can know it clearly its impact on crop yield.

4) In figure 1 A, the unit for chlorophyll content is not mentioned. The authors also need to mention seedling height plant -1 in Figure 1B.

5)In figure 2, the Y axis is Plant height should be cm plant -1  instead of cm. Or the authors also mention unit of chlorophyll content, straw biomass, number of panicles. Please double check it.

6)In figure 3, please mention the grain yield (g plant-1) instead of grain yield (g).

7)For Table 1 and 2, the unit of all parameters should be mention so that people can know it clearly.

8)  Scientific names in entire manuscript should be Italic.

8) Author should recheck the manuscript   and reference format according to instruction to author.

Author Response

Reviewer 2:

No.

Comments

Responses

1

Review report on manuscript (crops-2003820) entitled: Plant growth and yield response to salinity stress of rice grown under the application of different N levels and Bacillus pumilus strain TUAT-1. Paper is well written, results are clearly presented and discussed. Following comments should be addressed before further process.  

Thank you very much for the constructive comments. We agreed and revised the manuscript accordingly.

2

 1)     The English language should be thoroughly checked. There are many wrongly written sentences and phrases. Below few corrections are mentioned:

Thank you. We requested a native speaker for this but could not make an effort with professional English editing services.

3

a)      Line 49. space is required

Revised accordingly.

4

b)      Line 79. The reference mention in two type, reference number and author name, author should use only reference number instead of author name in manuscript text.

Followed the guideline and revised. L91

5

c)      Line 99-102. Expression of sentence is not clear in material method section.

We revised for more clarity. L113~116.

6

d)      Line 125.  Author should check unit of PAR of 500 mmol photons m-2s -1. It makes irrelevant data which is beyond the limit.

The unit is µmol m-2 s-1. We revised it. L144

7

e)      Line 144. Each replicate was measured 30 times, it is an unrealistic approach.

Yes, unrealistic, but we measured 30 times which is too much. This is because a spad meter takes a few seconds to take a one-time reading (about 2 seconds). It can continuously measure up to 30 values and can also calculate/display the average data in memory.

8

f)       Line 219.  Figures, Tables and Schemes heading should be avoided while writing the manuscript. Author should delete it accordingly.

We deleted accordingly. L244

9

2)Author  used   abbrebiation N in title, but what it is denotining is not clear enough. Author should work on the  title such that it clearly mention the research work use of abrevation must be minimised  and should be explained properly in title.

Thank you. We followed and revised it.

10

3)Authors   should mention the final recommendation of different N levels and  Bacillus pumilus strain TUAT-1 on crop yield in abstract section of manuscript so that people can know it clearly its impact on crop yield.

We revised and added the N rate in the brief conclusion in the abstract. L46~48.

11

4) In figure 1 A, the unit for chlorophyll content is not mentioned. The authors also need to mention seedling height plant -1 in Figure 1B.

We followed and revised.

12

5)In figure 2, the Y axis is Plant height should be cm plant -1  instead of cm. Or the authors also mention unit of chlorophyll content, straw biomass, number of panicles. Please double check it.

We followed and revised.

13

6)In figure 3, please mention the grain yield (g plant-1) instead of grain yield (g).

We followed and revised.

7)For Table 1 and 2, the unit of all parameters should be mention so that people can know it clearly.

We followed and revised.

8)  Scientific names in entire manuscript should be Italic.

We followed and revised.

8) Author should recheck the manuscript   and reference format according to instruction to author.

We rechecked and formatted the reference.

Round 2

Reviewer 1 Report (Previous Reviewer 1)

No more comments.

Reviewer 2 Report (Previous Reviewer 2)

Manuscript can be accepted

This manuscript is a resubmission of an earlier submission. The following is a list of the peer review reports and author responses from that submission.

Round 1

Reviewer 1 Report

Authors Win et al. conducted a greenhouse experiment to explore the impact of Bacillus pumilus strain TUAT-1 on growth and quality of rice under different nitrogen levels. Although, manuscript contains a lot of valuable information and data to enhance the productivity of rice crop but there are some flaws which are observed while review process. The following concerns are raised that may be addressed to improve the quality of draft before acceptance.

Abstract

Nitrogen levels are not mentioned in the abstract. Also interpret the percentage increase/decrease of the treatment with respect to control. Which parameters/indicators were under observation regarding salinity stress.

Line 30: these, mention the parameters.

Keywords: Enlist in alphabetically order.

Introduction

Line 40: factors.

Line 56: K+.

Line 65: remove 1.

Introduction is not properly managed. It must be improved with scientific approach. It contains a lot of irrelevant material. Write up looks poor.

Materials and methods

Line 72: 30 second.

Line 73: 2 minutes.

Line 74: 27℃.

Line 77: What do you meant by 0%, 100% and 150%. Not clear. Please elaborate accordingly.

Line 80: reference is missing.

Line 83: minutes.

Line 93: 30/25℃.

Line 97-98: Looks repetition.

Line 106: maturity; how many days after transplantation/conditions of maturity?

Line 105: 80℃.

Materials and method section needs improvement to ensure the repetition study. How salinity stress was imposed? Measurements of all parameters, discussed in results section, are not given. Data of stress indicators are not observed. It is a major drawback.

Results and Discussion

Results are not properly managed. Presentation looks poor. It must be improved accordingly. Discussion portion is too short and not explained with logical approaches. No mechanism is explained or discussed to reveal the defensive impact of nitrogen and/or TUAT-1.

Conclusion

It is too lengthy. Normally, conclusions are quantified and concise. Recommendation is missing.

Reviewer 2 Report

The manuscript entitled “Plant growth and yield response to salinity stress of rice grown under the application of different N levels and Bacillus pumilus strain TUAT-1” has done a study on investigating synergistic effects of N levels and Bacillus pumilus strain TUAT-1 bioinoculant application on the growth and yield of rice grown under salt stress. The study is very basic and could have been more interesting. 

A few comments are appended below to help the author to improve the article.   

Line 42-43: provide the latest detail

Line 54: “…….. managing of biotic and abiotic stresses……” Grammar error

Two keywords have been used in the manuscript: salinity tolerance and salt tolerance, Please use one

Line 65-67: objective(s) could be clearer 

Line 84: mL−c , Please correct

Line 97: The manufacturer’s detail can be provided for commercial Shinano soil 

Line 164-167: the author has cited three references (32-33) saying PGPR produce phytohormones and it helps in the development of the plant. None of them mention Bacillus pumilus. The author is suggested to cite the articles Bacillus pumilus producing plant hormones, it would be more justified 

Line 171: detoxify ROS is not a genuine phrase to use in a research manuscript

Since the manuscript is all about stress, hence it is expected to investigate the stress biomarkers like antioxidant enzymes to understand how TUAT-1 is helping in alleviating the salt stress

In table 1, a, b, and c have been used to signify the level of significance, but it has not been mentioned in the table heading

Provide figures in better resolution and captions of figure separately in the text.